# Electrochemical Determination of Hydroxyurea in a Complex Biological Matrix Using MoS_2_-Modified Electrodes and Chemometrics

**DOI:** 10.3390/biomedicines9010006

**Published:** 2020-12-24

**Authors:** Remi Cazelles, Rajendra P. Shukla, Russell E. Ware, Alexander A. Vinks, Hadar Ben-Yoav

**Affiliations:** 1Nanobioelectronics Laboratory (NBEL), Department of Biomedical Engineering, Ilse Katz Institute of Nanoscale Science and Technology, Ben-Gurion University of the Negev, Beer-Sheva 8410501, Israel; rajendra@post.bgu.ac.il; 2Division of Pediatric Hematology, Cincinnati Children’s Hospital Medical Center, Cincinnati, OH 45267, USA; Russell.Ware@cchmc.org; 3Global Health Center, Cincinnati Children’s Hospital Medical Center, Cincinnati, OH 45267, USA; 4Department of Pediatrics, College of Medicine, University of Cincinnati, Cincinnati, OH 45267, USA; sander.vinks@cchmc.org; 5Division of Clinical Pharmacology, Cincinnati Children’s Hospital Medical Center, Cincinnati, OH 45267, USA

**Keywords:** hydroxyurea, molybdenum sulfide, electroanalytical chemistry, chemometrics, electronic tongue, sickle cell disease

## Abstract

Hydroxyurea, an oral medication with important clinical benefits in the treatment of sickle cell anemia, can be accurately determined in plasma with a transition metal dichalcogenide-based electrochemical sensor. We used a two-dimensional molybdenum sulfide material (MoS_2_) selectively electrodeposited on a polycrystalline gold electrode via tailored waveform polarization in the gold electrical double layer formation region. The electro-activity of the modified electrode depends on the electrical waveform parameters used to electro-deposit MoS_2_. The concomitant oxidation of the MoS_2_ material during its electrodeposition allows for the tuning of the sensor’s specificity. Chemometrics, utilizing mathematical procedures such as principal component analysis and multivariable partial least square regression, were used to process the electrochemical data generated at the bare and the modified electrodes, thus allowing the hydroxyurea concentrations to be predicted in human plasma. A limit-of-detection of 22 nM and a sensitivity of 37 nA cm^−2^ µM^−1^ were found to be suitable for pharmaceutical and clinical applications.

## 1. Introduction

Sickle cell anemia is a common inherited blood disorder that leads to major morbidity and early mortality [1,2,3,4]. Hydroxyurea (HU) has been approved by both the United States Food and Drug Administration (FDA) as well as the European Medicines Agency [2]. Typical weight-based dosing with a stepwise escalation to the maximum tolerated dose (MTD) leads to predictable laboratory and clinical benefits; however, this often takes 6 to 12 months to achieve. In order to increase sickle cell treatment efficacy, it is recommended to use a pharmacokinetics (PK)-based HU dosing strategy [3,4], which involves accurately measuring HU in plasma or blood to generate individualized dosing recommendations [5,6].

A wide variety of analytical techniques have been developed and used to measure HU quantitatively in biological samples [7]. Chemical reactions with colorimetric endpoints are the time-honored technique, which were supplanted by chemical separation using high performance liquid chromatography (HPLC) or nuclear resonance (NMR). More recently, gas chromatography with mass spectrometry detection or tandem liquid mass spectrometry has been developed for the accurate measurement of the HU concentration in fluids. The current analytical techniques require expensive equipment and technical expertise [7,8]; therefore, establishing a simple, robust, and accurate point-of-care (POC) assay for HU as part of a PK-guided precision dosing algorithm [9] would represent a major step toward increasing the global use of this live-saving disease-modifying therapy [10,11].

Current standard methods for testing HU blood levels are based on chromatographic analysis, coupled with electrochemical detection or mass spectrometry [12,13,14]. The disadvantages of chromatography analysis for real-time and POC testing range from the use of costly machines requiring specialized staff, to the time-consuming manipulation of the sample, which is needed during the preparation steps. Since HU is electrochemically active [15], electrochemical sensors are optimal tools for quantifying HU. Interestingly, these sensors can be modified with selective coatings to provide built-in selectivity toward the molecule of interest [15,16,17,18]. An important challenge regarding measuring HU directly in blood samples is to deal with the interfering signals of other electro-active species present in the blood [14,19].

The electrochemical determination of low-molecular weight drugs such as HU (i.e., 76 g mol^−1^) dispersed in a complex biofluid such as blood or plasma often requires the constituents to be pre-separated by chromatographic pretreatment [12,13,20]. The voltammetric determination of HU usually occurs at a high polarization potential; however, this often leads to undesired side reactions within the surrounding matrix, which may decrease the lifetime of the sensor [15,21]. Voltammetric electronic tongues are bioinspired sensing units based on large dataset acquisition, followed by multivariate signal processing [22,23]. They consist of an array of sensors bearing different partial selectivities toward one or more molecules of interest (i.e., HU). Each sensor can generate a different electrochemical fingerprint of the sample under investigation for each electrode in the array. The cross reactivity of the sensor array is usually employed to discriminate between populations of different samples, and it is possible to increase the sensitivity of the analysis by accumulating more fingerprint patterns using various electrochemical techniques. The large quantity of data generated is analyzed using chemometric tools; principal component analysis (PCA) enables the number of variables in the dataset to be reduced, and partial least square regression (PLSR) analysis enables the concentration of a single analyte in a complex matrix to be estimated [22,24,25].

Among the novel sensing materials used for electrochemical applications, the two-dimensional materials have unique electronic, optical, and mechanical properties that have attracted much attention [26]. Thanks to their outstanding characteristics, although graphene is the gold standard, transition metal dichalcogenide materials (TMDCs) have great potential to serve as new materials for advanced electrochemical sensors [27,28]. Specifically, monolayers of molybdenum sulfide (MoS_2_) possess excellent characteristics, such as high electron mobility as well as tunable and highly sensitive surface-enhanced Raman scattering activity [29]. In addition, TMDC’s material properties can easily be tailored to improve the performance of the electrochemical sensor [30,31,32]. We used exotic electrical waveforms for the electrodeposition of MoS_2_, allowing the properties of the electrodeposited material to be fine-tuned [31,32,33]. We show that it is possible to refine, in situ, the analysis of unlabeled redox-active molecules in biofluids using a stable electrochemical sensor to induce better cross reactivity and a better prediction output (Figure 1).

We modified the sensing electrodes with a MoS_2_ monolayer selectively electrodeposited using two different electronic polarization waveforms in order to fine-tune the specific electronic properties of the TMDC material. This resulted in the electrochemical differentiation of HU from the main redox active interfering molecules contained in blood, uric (UA), and ascorbic (AA) acids. It was also possible to determine the HU concentration in complex molecular environments such as simulated and real human sera (Figure 1a) [34]. We have been using chemometric analysis [24,25,35], allowing us to quantify the drug HU with a sensitivity of 37 nM cm^−2^ µA^−1^ and a limit of detection (LoD) as low as 22 nM in undiluted human serum. This new technique is both novel and innovative, as it offers an opportunity to measure HU with expensive technology and potentially even at the point-of-care. The utility of this novel detection system will be realized as HU becomes more widely used in low-resource settings, such as in sub-Saharan Africa [36].

## 2. Experimental Section

### 2.1. Consumables and Equipment

Hydroxyurea (98%, H8627, Sigma-Aldrich, Ltd., Rehovot, Israel), molybdenum sulfite (GLMSW0A1, ACS Material LLC, Pasadena, CA, United States), uric acid (≥99%, 01935, CHEM IMPEX, Inc., Wood Dale, IL, United States), ascorbic acid (>99.5%, BIA0602, Apollo Scientific Ltd., Cheshire, United Kingdom), D-glucose (99%, A16828, Alfa Aesar, Lancashire, United Kingdom), dopamine (99%, A11136, Alfa Aesar), methanol (>99.8%, 001368052100, Bio-Lab, Ltd., Jerusalem, Israel), L-homocysteine (BIB6065, Apollo Scientific, Ltd.), magnesium sulfate (>99%, 931255, STREM Chemicals, Inc., Newburyport, MA, United States), ammonium carbonate (1716477, Fisher Scientific, Hampton, NH, United States), calcium chloride (97%, 10195054 Alfa Aesar), iron sulfate hydrate (97%, 307718, Sigma-Aldrich), hydrochloric acid (32%, 00846050100, Bio-Lab Ltd.), sulfuric acid (95–98%, 001955050100, Bio-Lab, Ltd.), phosphoric acid (85%, 65324100, Daejung Chemicals & Metals Co., Ltd., Shiheung-city, China), potassium permanganate (≥99%, 36675, Alfa Aesar), disodium hydrogen phosphate dihydrate (≥99.5%, 1.06580.1000, Merck, Darmstadt, Germany), sodium dihydrogen phosphate dehydrate (≥99%, 1.06342.0250, Merck), sodium chloride (≥99.5%, 1259991, Merck), potassium hexacyanoferrate (II) trihydrate (‘ferrocyanide’, 99%, 1.04984.0100, Merck), potassium hexacyanoferrate (III) (‘ferricyanide’, ≥99.0%, 1.04973.0100, Merck), acetone (99.8%, 010376, Bio-Lab, Ltd.), hydrogen peroxide (30%, 1.07210.1000, Merck), potassium chloride (99%, 11595, Alfa Aesar), and 2-propanol (>99.8%, 1301221, Bio-Lab, Ltd.) were used without further purification. Deionized water was obtained from a Super Q water system (Millipore system, resistivity > 18 MΩ, Merck). Origin^TM^ Pro software (2018, SR1 edition, Northampton, MA, United States) was used for all data analyses and figure plots.

### 2.2. Buffered, Simulated Serum, and Serum Sample Preparation

The buffer and simulated serum were prepared using the crystalline or liquid forms of the commercial products and were used without further purification. Phosphate-buffered saline was prepared as a 10× stock solution and was diluted daily for the purpose of the experiments in this paper. The 10× PBS pH 7.4 solution was prepared by adding 1.37 M NaCl (40 g), 27 mM KCl (1 g), 1 M Na_2_HPO_4_ (7.2 g), and 18 mM KH_2_PO_4_ (1.2 g) in MQ water (0.5 L). A ferrocyanide/ferricyanide electrochemical characterization solution was prepared with an equimolar quantity of ferrocyanide (105.5 mg) and ferricyanide (82.5 mg) dispersed in 1× PBS solution (100 mL). HU standard solutions of 50, 100, 150, 200, 250, 300, 350, 400, 450, and 500 µM were prepared daily by successive dilution of a 10 mM HU (15.2 mg) stock solution (20 mL).

The simulated serum was prepared by adding each salt in MQ water under stirring at 600 rpm (800 mL). Each compound concentration lies in its maximum biological concentration in order to exacerbate the synergic effects, the detailed simulated serum composition is described in Appendix A. Serum samples were collected from a 38-year-old healthy volunteer (15 mL tubes, BD Vacutainer^®^ SSTTM II Advance, Fisher Scientific, Ltd.); blood is let to clot at room temperature for 30 min. The clotted blood is centrifuged (1200 rpm for 10 min) and the supernatant is collected and stored in new tubes. The supernatant is collected following another centrifugation step (1200 rpm for 10 min) and aliquots (0.5 mL) are stored in Eppendorf tubes at −20 °C. The trial was approved and registered by the Human Research Ethics Committee of the Ben-Gurion University of the Negev, Israel (#1601-2, 10 September 2018). Written informed consent was obtained from all participants.

### 2.3. Electrodeposition of MoS_2_

We used a VSP potentiostat (Bio-Logic, Ltd., Seyssinet-Pariset, France) for the electrodeposition and the characterization of the electrodes. A three-electrode cell configuration was used, consisting of a commercial gold electrode (932-00023, Gamry; working electrode; ‘WE’ with a surface area of 0.07 cm^2^), an externally applied commercial Pt wire with an approximate surface area of 3.6 cm^2^ (CHI115, CH Instruments; counter electrode; ‘CE’), and an Ag/AgCl 3 M NaCl reference electrode (CHI111, CH Instruments; reference electrode; ‘RE’, E_SHE_ = 0.210 + E_Ag/AgCl_). All electrochemical potential values are versus Ag/AgCl half-cell potential. The WE was dipped in 1 g L^−1^ MoS_2_ solution dispersed in 0.1 M sulfuric acid (10 mL). The material was selectively deposited using the CV technique. The first sample was deposited according to a previously published procedure by cycling the potential 10 times at 50 mV s^−1^ between −1 and 1 V_Ag/AgCl_ [19]. The other samples were deposited in the same time window (800 s) using a potential range corresponding to gold electrochemical double layer formation region (EDL, −0.3–0.7 V_Ag/AgCl_) at 50 mV s^−1^ for 20 cycles, 1 V s^−1^ for 400 cycles, and at 10 V s^−1^ for 4000 cycles. The last sample was deposited in an extended potential range (Extended EDL, 0–1.4 V_Ag/AgCl_) at 1 V s^−1^ for 400 cycles. Prior to the electrodeposition, the commercial gold electrodes were polished with 0.3 and 0.05 µm alumina slurry and subsequently sonicated in MQ water. The electrodes were further electro-chemically cleaned using CV in a 0.5 M H_2_SO_4_ electrolyte (10 mL) by cycling the potential from 1.0 V to 1.5 V and back to −0.4 V for 5 to 10 cycles until a steady voltammogram representative of a clean substrate was obtained (note that it is often necessary to renew the H_2_SO_4_ solution).

### 2.4. Electrochemical Characterization

The electrochemical activity of the bare gold and the MoS_2_-modified electrodes was tested in the presence of 5 mM ferrocyanide/ferricyanide solution (20 mL) using CV at 100 mV s^−1^ in the potential range −0.2–0.6 V_Ag/AgCl_. The electrochemical impedance spectroscopy (EIS) measurements were subsequently recorded at an open circuit potential (0.20 ± 0.05 V_Ag/AgCl_) from 10 kHz to 0.1 Hz with eight points per decade and with a sine-wave amplitude of 12 mV. The fitting of the EIS spectra was done via BioLogic EC-Lab software (version 11.36, Seyssinet-Pariset, France); two different equivalent electronic circuits were used for the bare and the MoS_2_-modified electrodes (Appendix A).

### 2.5. Electrochemical Sensing of Hydroxyurea, Uric Acid, and L-Ascorbic Acid in PBS and Simulated Serum

HU containing PBS, simulated and real human serum solutions (5 mL) were characterized by differential pulse voltammetry (DPV) on a Biologic VSP potentiostat. The initial potential was −0.2 V, the vertex potential was 0.6 V, the pulse height was 0.1 V, the pulse width was 0.01 s, the step height was 5 mV, and the step time was 0.1 s (equivalent scan rate: 50 mV s^−1^). The calibration curve of HU, UA, and AA in PBS were acquired within their pharmaceutical and biological relevant concentrations ranges (0–500 µM, 0–400 µM, and 0–250 µM, respectively). Triplicates were recorded and averaged using the multiple curve average tool available from Origin^TM^ Pro software (2018, SR1 edition, Northampton, MA, United States). The sensitivity was calculated as the slope of the linear regression analysis plot; the errors bars for the sensitivity were determined by linear regression analysis. We followed the IUPAC guidelines to calculate the limit of detection (LoD), taken as three times the intercept error divided by the slope with the error on the limit of detection calculated as the intercept error divided by the slope, a method used for comparison purpose [37,38].

The simultaneous analysis of HU samples in undiluted human serum with three different electrodes was carried out with a multi-channel potentiostat (CompactStat.h; Ivium Technologies B.V., Eindhoven, Netherlands). The serum samples were spiked with 20× HU stock concentrations (1, 2, 3, 4, 5, 6, 7, 8, 9, and 10 mM in PBS) and the three electrochemical analyses were then carried out (DPV, CV, and CA). DPV analysis conditions were the same as described earlier (E_ini_ = −0.2 V, E_fin_ = 0.6 V, pulse height = 0.1 V, pulse width = 0.01 s, step height = 5 mV, step time = 0.1 s). The CV analysis was carried out at 500 V s^−1^ in the potential range of −0.2 to 0.6 V_Ag/AgCl_ and the CA analysis was carried out at seven different potentials (−0.2, −0.1, 0, 0.1, 0.2, 0.3, 0.4, 0.5, and 0.6 V_Ag/AgCl_) for 0.5 s with a sampling of 0.01 s.

### 2.6. Chemometric Analysis

Origin software (OriginPro 2018 (64-bit) SR1; b9.5.1.195, Northampton, MA, United States) was used to perform the linear regression analysis, allowing the extraction of the slopes (e.g., sensitivities) and the intercepts (e.g., to calculate the LoD) from the calibration datasets. Multivariate analyses such as PCA and PLSR analysis were carried out according to the journal’s guidelines. Smoothing of raw data was performed using the Savitzky-Golay method with 32 points of window, no boundary conditions, and with polynomial order 2. We used the cubic spline interpolation method to reduce the 160 datapoints of the DPV analysis in human serum to the 17 specific potentials (−0.2, −0.15, −0.1, −0.05, 0, 0.05, 0.1, 0.15, 0.2, 0.25, 0.3, 0.35, 0.4, 0.45, 0.5, 0.55, and 0.6 V_Ag/AgCl_).

## 3. Results and Discussion

### 3.1. Electrodeposition of MoS_2_ on Polycrystalline Gold Electrodes

We electrodeposited MoS_2_ on polycrystalline gold electrodes for the electrochemical determination of HU using multivariate analysis models (Figure 1a) [25]. Electrodeposition of MoS_2_ allows for the fine-tuning of the faradaic and capacitive properties of the TMDC-modified electrode, which is possible with a tailored waveform polarization (Figure 1b). The deposition of MoS_2_ on the gold substrate was confirmed by scanning electron microscopy and energy-dispersive X-ray spectroscopy analysis (Appendix A); the elemental composition corresponds to a molar composition of two sulfur atoms per molybdenum atom (Mo_1_S_2.02_). The MoS_2_ material was electrodeposited with cyclic voltammetry (CV), as reported earlier (Figure 1b, grey dashed trace) [19], and by using a smaller potential window limited to the gold electrochemical double layer (EDL) formation region. The black traces correspond to the electrodeposition of the TMDC material in the EDL region at different scan rates. The gold EDL region exists between −0.3 and 0.7 V_Ag/AgCl_ [39]. At these potentials the pseudo-capacitive currents emerging from the reversible adsorption of the electrolytes’ ion dominate. The potential window for the electrodeposition of MoS_2_ was later increased (Figure 1b, grey dotted trace), leading to the concomitant formation of a gold hydroxide adduct (+0.8 V_Ag/AgCl_) and the formation of a gold oxide layer (+1.3 V_Ag/AgCl_). The deposition of MoS_2_ with CV at high polarization potentials, as reported earlier (−1–1 V_Ag/AgCl_) [19], involves the consecutive oxidation/reduction of molybdenum atoms within the lattice (E(Mo^IV^/Mo^VI^) = 0.7 V_Ag/AgCl_) [34], and the creation of sulfur defects at E < −0.5 V_Ag/AgCl_ (Figure 1b) [40]. As a result, the quantity of charge transferred during the electrodeposition is negative, with an increase in the reduction peak at −0.5 V_Ag/AgCl_ (Appendix A). When MoS_2_ is deposited within the gold’s EDL, the quantity of charge transferred is positive and the cyclic voltammograms show a gradual decrease in the capacitive currents, whereas the oxidation wave at 1.3 V_Ag/AgCl_ is increased when the extended EDL electrochemical windows is used (Appendix A). The cyclic voltammogram of 10 mg L^−1^ MoS_2_ in 0.1 M sulfuric acid solution displays three oxidation waves (0.8, 1.4, and 1.6 V_Ag/AgCl_) associated with the piecewise oxidation of the lattice Mo atom [34], -SS-, and Mo-S^−^ bond oxidation as well as the total conversion of MoS_2_ to MoO_3_ (Appendix A) [41,42,43,44,45]. The creation of partially oxidized Mo sites for a deposition potential >1.0 V_Ag/AgCl_ leads to increased charge being transferred (Q–Q_0_ in Coulomb, Appendix A). The creation of defects on the surface of the electrodeposited MoS_2_ allows one to increase the number of catalytic sites or the number of defects in the material’s 2D structure [44]; however, it decreases the stability of the electrodeposited material at the nanoscale [43].

We used CV and electrochemical impedance spectroscopy (EIS), in the presence of ferrocyanide/ferricyanide, as a redox probe to characterize the faradaic and capacitive features of the gold electrodes modified with MoS_2_. The bare electrode displays the narrowest peak-to-peak potential with the highest current density (Figure 1c, yellow, *I_anodic_* = *I_cathodic_* = 0.28 mA cm^−2^, ∆*E* = 66.6 mV vs. 59 mV for fully reversible systems). For electrodes modified with Sun H. et al. protocol (CV between −1/+1 V_Ag/AgCl_, dotted gray trace) [19], we observed a small decrease in anodic and cathodic currents compared to those at the bare electrode, which correlates with the observations shown in Figure 1d (i.e., (i) the decrease of capacitive currents or (ii) the anodic oxidation peak shift of UA in the differential pulse voltammetry (DPV) analysis in the presence of 400 µM UA). For the gold EDL-based electrodepositions, the effective modification of MoS_2_ induces a gradual decrease in the peaks’ current densities for both the anodic and cathodic peaks (*I_anodic_* = *I_cathodic_* = 0.26, 0.14 and 0.03 mA cm^−2^ for deposition at 0.05, 1 and 10 V s^−1^, respectively) and an increase in the peak-to-peak potentials (Figure 1c, ∆*E* = 150 mV at 50 mV s^−1^, ∆*E* = 450 mV at 1 V s^−1^). The specific capacitance and resistance values of each of the electrodes are calculated by fitting of the Nyquist plots of the EIS analysis (Appendix A). To increase the number of cycles via increasing the scan rate leads to an increase in electron transfer resistance by 146-fold (from 175 Ω for bare gold to 25,700 Ω for two resistances added in a series and corresponding to the gold and to the MoS_2_ layer). The decrease of the capacitance observed by EIS (from 0.6 to 0.2 µF), due to the MoS_2_ add-layer, is consistent with the decrease of capacitive current observed in the DPV analysis in the presence of UA, chosen as a natural redox active analyte (Figure 1d). The decrease of capacitance in the range −0.2 to 0.1 V_Ag/AgCl_ is more pronounced for those samples modified at higher scan rates (having a greater number of deposition cycles). The shift in the UA oxidation peak is correlated with the capacitance change in the gold electrical double layer (*C_EDL_*, see Appendix A). A higher capacitance (*C_EDL_* > 0.6 µF) shifts the oxidation peak of UA toward a more oxidative potential (∆*E* = 80 mV), and a lower capacitance shift UA oxidation peak shifts it to lower potentials (∆*E* = 230 mV for *C_EDL_* = 0.2 µF). The modification of the electrode using the protocol published by Sun H et al. only slightly affects the oxidation peak of UA; the electro deposition of MoS_2_ at 10 V s^−1^ allows for the decreasing of the oxidation potential of UA by 230 mV. The electrodeposition of MoS_2_ within the gold EDL region is more effective at high scan rates due to the larger number of polarization cycles. By combining both the CV and EIS results, we noted that a large electrochemical window involving a complex redox reaction for both the material (MoS_2_/MoS, Mo^IV/VI^) and the polycrystalline gold substrate (Au/Au-OH, Au^I/III^) also leads to the deposition of the MoS_2_ material. The decrease in the capacitive current in the range of −0.2 to 0.1 V_Ag/AgCl_, observed in the DPV analysis for the MoS_2_-modified electrodes, are correlated with the decrease in *C_EDL_* obtained from the EIS analysis.

### 3.2. Electrochemical Signatures of Hydroxyurea, Uric Acid, and Ascorbic Acid

The electrochemical signature of HU, on a bare polycrystalline gold electrode and in phosphate-buffered saline (PBS) at pH 7.4 (Figure 2a), presents a positive dose-response peak at 0.1 V_Ag/AgCl_, attributed to the oxidation of HU, and a negative dose-response feature centered at 0.4 V_Ag/AgCl_, attributed to the decrease of the electrode pseudo-capacitance. The background voltammogram recorded at the clean bare gold electrode presents two pseudo-capacitive waves centered at −0.2 V_Ag/AgCl_ and 0.4 V_Ag/AgCl_ due to the adsorption of the buffer’s anions at gold adatom sites (c.a. phosphate and chlorine) [46]. Gold adatoms are inherently present on commercial polycrystalline gold electrodes; they present enhanced electrochemical activity due to their partial oxidation state, namely, Au^I^_ads_ [47]. The electrochemical determination of HU at low potentials is possibly due to its concomitant adsorption and oxidation following a sequential path (Equations (1)–(3)) [15].
H_2_NCONHOH → H_2_NCONO(ads) + 2 H^+^ + 2 e^−^(1)
H_2_NCONO(ads) → HNCONO(ads) + H^+^ + e^−^(2)
HNCONO(ads) → 2 NCO + H^+^ + e^−^(3)

Upon the addition of HU (ca. 50 µM), the pseudo-capacitive current observed at the bare gold between 0.1 and 0.6 V_Ag/AgCl_ vanishes, due to the irreversible adsorption of HU oxidation products (Equation (1)), hindering the reversible adsorption of the chlorine buffer’s anions. The further oxidation of HU on gold (E_Ox1_ ≥ 0.25 V_Ag/AgCl_) is linked to the formation of new partially oxidized adsorbate species [15], thus explaining the dose response-dependent decrease of a pseudo-capacitive currents centered at 0.4 V_Ag/AgCl_ with an increasing HU concentration (Figure 2a).

The MoS_2_-modified electrode allows the first oxidation step of HU to be decreased by 300 mV (Figure 2b). The DPV presents both a capacitive and faradaic positive dose-response distributed over the full (−0.2–0.6 V_Ag/AgCl_) potential range. The background voltammogram acquired in a PBS solution pH 7.4 (Figure 2b, black trace) is linear until the onset of lattice molybdenum atoms oxidation (E_on_ = 0.5 V_Ag/AgCl_) [34]. The pseudo-capacitive wave observed at the bare gold electrode and linked to the buffer ions’ reversible adsorption is completely suppressed at the MoS_2_-modified electrode. Instead, capacitive current over an extended potential range emerges upon the addition of HU (ca. 50 µM). The oxidation mechanism underlying HU determination is ambiguous at concentrations below 150 µM until a clear oxidation pattern is centered at −0.2, 0.1, and 0.6 V_Ag/AgCl_. It emerges with similar dose response sensitivities of 35 ± 3 mA cm^−2^ M^−1^. The two first oxidation peaks, located at −0.2 V_Ag/AgCl_ and 0.1 V_Ag/AgCl_, correspond to the partial oxidation of HU (Equations (1) and (2)). The second peak’s duplicity tends to disappear at high concentrations (e.g., >350 µM); this is attributed to the absorption of oxidation byproducts. The total oxidation of HU on the MoS_2_-modified gold occurs at 0.6 V_Ag/AgCl_, which is 200 mV lower than that of a bare electrode [21]. It corresponds to the concomitant deprotonation and oxidation of the adsorbed oxidation by-products (Equation (3)) [15]. The ability of the MoS_2_-modified gold electrode to sense HU at a potential <0 V_Ag/AgCl_ is most likely associated with the hydronium adsorption properties of TMDC materials, favoring the electrochemical oxidation deprotonation step [34].

We calculated the sensitivity of each electrode construct from linear regression analysis over the full DPV potential range (Figure 2c). Capacitive and faradaic currents contribute synergistically to the dose-response features of each electrode; the sensitivities are 44.9 ± 1.7 mA cm^−2^ M^−1^ and 35.5 ± 0.4 mA cm^−2^ M^−1^ at 0.1 V_Ag/AgCl_ for bare gold and MoS_2_ electrodes, respectively. The MoS_2_ electrode presents two supplementary features at −0.2 and 0.6 V_Ag/AgCl_ with a sensitivity of 35 ± 6 mA cm^−2^ M^−1^ and 37 ± 2 mA cm^−2^ M^−1^, respectively. The coefficient of determination resulting from linear regression at those potentials is >0.9 (Appendix A). Given the opportunity to quantify HU on a broad range of potentials, we defined the combined sensitivity (*CS*) of the DPV analysis for a given electrode in the potential range (Equation (4)).
*CS_DPV_*{−0.2 < V_Ag/AgCl_ < 0.6} = ∑*S_V_*, for R^2^_(Sn)_ > 0.9(4)

A higher *CS_DPV_* value correlates with a larger number of unique potentials at which the molecule of interest can be accurately analyzed. Electrode constructs with a high *CS_DPV_* value perform better in estimating an analyte concentration using multivariate calibration methods. The *CS* of the DPV analysis for bare gold (*CS_DPV_* = 101 ± 2 mA cm^−2^ nM^−1^) is substantially lower than that of the MoS_2_-modified electrode (*CS_DPV_* = 199 ± 3 mA cm^−2^ nM^−1^), confirming the advantages of the TMDC in determining the HU concentration in PBS. The LoD for HU (Figure 2d) at the MoS_2_-modified electrode (LoD_(E = 0.1V)_ = 9.4 ± 3.1 µM) is improved, compared to the bare gold electrode (LoD_(E = 0.1V)_ = 28.8 ± 9.6 µM) (Appendix A). Whereas the bare gold electrode fails in accurately quantifying HU below its biological range (Limit of quantification = 3 × LoD > 50 µM), the MoS_2_-modified electrode allows to accurately quantify HU within its biological concentration range at two different potentials, with thresholds as low as for HPLC analytical techniques (10–100 µM, Figure 2d) [12,13,20].

We investigated the specificity of the electrochemical sensors toward the oxidation of two major interfering species consistently encountered in biofluids (Figure 3). Redox active molecules such as UA [48] and AA [49] may generate electrochemical signals that mask the signature of HU. The electrodeposited MoS_2_ allows decreasing the oxidation peak potential of UA from 0.45 to 0.25 V_Ag/AgCl_ (Figure 3a). A smaller half peak width at the MoS_2_ electrode (112 mV vs. 168 mV at the bare gold) indicates a greater electrochemical reversibility of UA oxidation due to the enhanced mass transport [50]. The LoD values are above the biological concentration range of the analyte with the bare electrode and are within the biological concentration range of the analyte for the MoS_2_-modified electrode (Appendix A). The oxidation peak of AA is shifted to a lower potential at the MoS_2_ electrode (from 0.35 to 0.23 V_Ag/AgCl_). In addition to the decrease of capacitive currents from 10 to 3 µA cm^−2^ at 0 V_Ag/AgCl_, the modification of the electrode slightly widens the electrochemical signature of AA (Figure 3b).

We calculated the sensitivity of each electrode toward UA and AA over the potential range of −0.2 to 0.6 V_Ag/AgCl_ at increasing concentrations of analytes in PBS at pH 7.4 (Figure 3c). The electrodeposited MoS_2_ allows the electrochemical sensitivity to be increased toward UA oxidation by two-fold (from 49 ± 3 to 93 ± 5 mA cm^−2^ M^−1^), while slightly increasing the sensitivity toward AA at 0.4 V_Ag/ACl_ (from 17 ± 2 to 20 ± 0 mA cm^−2^ M^−1^). It is possible to increase the sensitivity toward UA seven-fold at 0.3 V_Ag/AgCl_ and 3.5-fold toward AA at 0.1 V_Ag/AgCl_ with the MoS_2_ modified electrode (Appendix A). We calculated the cumulative selectivity of those DPV analyses (*CS_DPV_*), emphasizing that the MoS_2_-modified electrode generates twice as many features for the electrochemical determination of HU and AA (Figure 3d). The *CS_DPV_* toward AA increases from 93 ± 2 to 204 ± 1 mA cm^−2^ M^−1^ and from 184 ± 3 to 216 ± 5 mA cm^−2^ M^−1^ toward UA. The decrease of the oxidation potential for the three analytes and the two-fold increase of sensitivity for UA on the MoS_2_ modified electrode may be explained by the relative hydrophobicity of the analytes (LogP_UA_ < LogP_HU_ < LogP_AA_ < LogP_H2O_, see Appendix A) favoring Van der Walls interactions with the electrodeposited material [51]. The sensitivity and limit of detection for HU is comparable to other MoS_2_ modified electrodes present in the scientific literature, and the sensitivity and limit of detection toward UA and AA falls in the same range as previously published results for different electrode materials (Appendix A).

### 3.3. Analysis of Hydroxyurea in a Simulated Serum Using Chemometrics

We prepared a simulated human serum matrix comprising 28 interfering molecules in their upper physiological range [14] buffered at physiological pH (Appendix A). The synthetic buffer used not only comprise the commonly encountered interfering molecules AA and UA but also 26 other redox active components susceptible to disrupt the electrochemical determination of HU. The background voltammogram of the simulated serum, recorded with the MoS_2_-modified electrode (Figure 4a, plain black trace), presents a major oxidation peak at the same potential of UA in PBS (29 µA cm^−2^ at 0.25 V_Ag/AgCl_), generating roughly 3/4 of the current density of that of a 400 µM UA solution (see Figure 3a). At the bare electrode, the simulated serum produces a similar oxidation peak that is shifted by 100 mV toward a cathodic potential, compared to the electrochemical signature of 400 µM UA in PBS (Appendix A).

The evolution of the DPV traces at an increasing HU concentration is rather complex in the presence of biological concentrations of UA, AA and 26 other redox active molecules (Figure 4a). The peak potential corresponding to UA oxidation is shifted to anodic values and pseudo-capacitive current increases at a low potential (−0.2–0.1 V_Ag/AgCl_), indicating a change of the interfacial equilibrium between the electrode, HU and the 28 molecules composing the simulated serum. At HU concentrations > 150 µM, the DPV traces are characterized by an increase in the peak current located between 0.4 and 0.6 V_Ag/AgCl_. The linear regression analysis, and *CS_DPV_* analysis of calibration curves over the full potential range, confirms the advantage of the MoS_2_ modified electrode in determining HU in a synthetic serum (LoD_E = 0.5V_ = 13 ± 4 µM and *CS_DPV_* = 102 ± 4 mA cm^−2^ M^−1^ for MoS_2_ vs. LoD_E = 0.5V_ = 24 ± 8 µM and *CS_DPV_* = 89 ± 2 mA cm^−2^ M^−1^ for the bare gold electrode).

Among chemometric algorithms, PCA can be used to condense large DPV datasets into a smaller set of new composite dimensions. The PCA can explain the variance-covariance structure of the HU calibrations curves over the full range of potentials, through a linear combination (the principal components, PC1 and PC2) that provide a maximized variance for the dataset (Figure 4b, MoS_2_-modified electrode in black; bare gold in light gray). The PCA exemplifies the piecewise variation of the DPV traces centered at 150 µM of HU. The loading plots (i.e., the red vector in Figure 4b) emphasize the positive dose-response relationship at 0.35 V_Ag/AgCl_, and the inverse dose response correlation at 0.25 V_Ag/AgCl_ at low HU concentration (0–150 µM), whereas this tendency is inverted for the bare gold electrode. For HU > 150 µM, it is the current variations in the 0.45–0.55 V_Ag/AgCl_ range that correlate directly with the increased HU concentrations for both electrodes.

Partial least squares regression analysis (PLSR) combines the features of PCA and multiple regression analysis and allow for the predicting of HU concentrations using the decomposition of the DPV’s current vs. potential variables. The variable importance plot (VIP) is a measure of the most significant variables that contribute to the DPVs’ profile variation for different HU concentrations (Figure 4c). The variables with scores >0.8 are considered important and may be chosen to generate the multivariate model, whose X and Y axes are a linear combination of current vs. potential vectors [35,52]. The MoS_2_-modified electrode accumulates 80 variables with VIP scores > 0.8 centered around two sets of potentials at 0.25 and 0.5 V_Ag/AgCl_, whereas the bare gold electrode’s 43 variables are centered at 0.55 V_Ag/AgCl_. We curated the original dataset from 202 to 17 main variables for computational economy needs, the two VIP analysis results superpose (Figure 4c), the VIP profiles for low and high HU concentrations differs significantly (Appendix A).

We used the PLSR analysis to predict the HU concentration over its full clinical range (0–500 µM, 0–37 ng mL^−1^), in PBS solution and in a simulated serum composed of 28 interfering molecules at their biological concentration (Figure 4d). The best multivariate model comprises the dataset of both electrodes, and a limit of detection 5.5 ± 1.8 µM is calculated (taken as three times the intercept error, Appendix A) [37,38]. The multivariate model uses the current vs. potential relationship at 0.55 V_Ag/AgCl_ for the gold electrode and at 0.45 and 0.5 V_Ag/AgCl_ for the MoS_2_ modified electrode; those variables present the highest VIP scores (2.04, 2.06 and 2.06, respectively). The limit of detection is lower than any univariate regression models (Appendix A). We use the predicted residual error sum of squares (PRESS) as a statistical measure of model accuracy [53]. The number of variables used by the model depends on the combination of the datasets. Due to a favorable synergistic effect, the calculated limit of detection decreases by two-fold in the simulated serum compared to the PLSR analysis in PBS only.

### 3.4. Analysis of Hydroxyurea in Real Human Serum with an Electrode Array

We combined three electrodes and three electrochemical technics to predict HU concentrations in human serum. We used a bare polycrystalline gold electrode, and two other electrodes that are modified with MoS_2_ (see Figure 1b, *ψ*_a_ corresponds to potential cycling at 1 V s^−1^ in the gold double layer region, and *ψ*_b_ involves the concomitant oxidation of MoS_2_ due to potential cycling at E > 0.8 V_Ag/AgCl_). The pseudo-capacitive current densities observed in the DPV analysis at 0 V_Ag/AgCl_ decrease after modification with MoS_2_ (I_MoS2-_*_ψ_*_a_ = 1.8 µA cm^−2^ < I_MoS2-_*_ψ_*_b_ = 6.0 µA cm^−2^ < I_Gold_ = 9.2 µA cm^−2^) and a cathodic potential shift of the main oxidation peak is observed when the TMDC material is deposited with *ψ*_b_ (E_MoS2-b_ = 0.45 V_Ag/AgCl_ < E_MoS2-a_ = E_Gold_ = 0.55 V_Ag/AgCl_, Figure 5a). The simultaneous acquisition of three electrochemical signals for the analysis of undiluted serum samples inherently generates an electrochemical noise due to the presence of >289 plasma proteins [54]. We reduced the electrochemical noise by smoothing and we curated the dataset to 17 main variables by extrapolation, as we know that does not change the result of the PLSR analysis (Figure 5a). The PCA analysis of the datasets before and after data curation display the same piecewise variation for low and high HU concentrations (Appendix A); hence, the synergic mixture of the redox active component present in the undiluted serum induces a piecewise variation of the datasets, which is adequately reproduced by our simulated serum.

We calculated the LoD for HU using either DPV, CV or CA for each of the electrode from univariate linear regression analysis with R^2^ > 0.9 (Figure 5b). The cumulative sensitivity parameters for each technic (*CS_Technic_*) are in the same order of magnitude as those calculated for the simulated serum. The lowest achievable LoD from univariate regression analysis is 20 ± 5 µM at the MoS_2_^b^ electrode using the DPV analysis with a sensitivity of 36.1 ± 0.8 mA cm^−2^ M^−1^. Those values are also comparable to those in the simulated serum (Figure 4d) and are suitable for clinical applications (ca. LoD < 50 µM).

The PCA analysis carried out using the whole dataset comprising three electrodes and three electrochemical technics also display the characteristic piecewise variation for low and high HU concentrations (Figure 5c). The vector parallel to the 200–500 µM HU concentration range in the loading plot correspond to the current variation of the DPV analysis carried out with the MoS_2_^b^ electrode (Figure 5c, red plot). The vectors corresponding to the CV analysis using MoS_2_^a^ and CA analysis using the bare gold electrode are also enlightened, and they belong to the 10 variables with highest VIP scores, allowing for an explanation of the variance of the HU calibration dataset in human serum.

Multivariate analysis was subsequently used to calculate the LoD value for HU in human serum with different electrode arrays and electroanalytical techniques array (Figure 5d). The PLSR model with the whole dataset takes advantage of each analysis and electrode constructs (LoD = 2.06 ± 0.69 µM, PRESS = 0.11 µM). The electroanalytical array is more sensitive (LoD = 0.41 ± 0.14 µM for the MoS_2_^a^ electrode, PRESS = 0.10 µM), while the electrode array is more accurate (lowest PRESS = 0.07 µM for DPV analysis, LoD = 7.8 ± 2.6 µM). The other two electrode arrays and two electroanalytical arrays are less performant (Figure 5d, large circles). Surprisingly, the array configurations that present the lowest LoD (22 ± 7 nM, composed of the DPV technic with the Gold and the MoS_2_^b^ electrode) exclude the variables corresponding to the most sensitive electroanalytical array (ca. the MoS_2_^a^ electrode). It is complicated to assess the sensitivity of such a multivariate regression method [35]; we use the combined selectivity of the PLSR analysis (*CS_PLSR_*) as a tool to compare the performance of each array. The *CS_PLSR_* corresponds to the average of sensitivities taken from univariate analysis and chosen from the variables selected by the multivariate model (Figure 5d). While the univariate analysis shows that the MoS_2_^b^ electrode is more sensitive, and correlate with highest *CS_PLSR_* values, the only array configuration that present both low LoD and high *CS_PLSR_* correspond to the electroanalytical array using the MoS_2_^a^ electrode construct (LoD = 0.41 ± 0.14 µM, *CS_PLSR_* = 29.5 ± 1.7 mA cm^−2^ M^−1^). The array configuration with the lowest LoD is relatively less sensitive; CV and CA datasets acquired with the MoS_2_^a^ electrode (LoD = 53 ± 17 nM with *CS_PSVR_* = 22.6 ± 1.3 mA cm^−2^ M^−1^), Gold and MoS_2_^b^ datasets resulting from DPV analysis (LoD = 22 ± 7 nM with *CS_PSV_* = 13.7 ± 0.6 mA cm^−2^ M^−1^) owning a maximum sensitivity from the DPV variables acquired at 0.55V with the MoS_2_^b^ electrode (S = 37 mA cm^−2^ M^−1^). Test analysis was carried out for two datasets in Figure 5d (CV + CA analysis with MoS_2_^a^ and DPV analysis of gold + MoS_2_^b^) corresponding to the lowest achievable LoD values. Each HU concentration is calculated with the multivariable linear regression model from the dataset excluding the tested HU concentrations (Figure 6). The linear regression analysis of predicted vs. actual HU concentration shows that both models are accurate (R^2^ = 0.99).

## 4. Conclusions

The present work quantifies HU, a life-saving therapeutic medication for sickle cell anemia, in human serum. It does so with great accuracy due to its unique electrochemical redox design, coupled with chemometrics. The selective electrodeposition of MoS_2_ on a gold electrode was carried out via electrochemical processes in the gold electrochemical double layer region. The chemisorption of the TMDC material via its sulfur moieties increased the electrochemical selectivity of HU and the two major redox active interfering species present in biofluids, namely UA and AA. The nature of the electronic waveform used for the electrodeposition induces in-situ structural changes in the TMDC material, generating additional specificity in the modified electrode. The current-to-HU concentration profile follows a simple linear relationship in commonly used PBS buffer, whereas a stepwise variation is observed in a complex matrix (i.e., a simulated human serum). An identical stepwise variation of the electrochemical signal is observed for HU analysis in actual undiluted human serum. It is possible to quantify HU in the clinical range (e.g., 50–500 µM). The combination of electrochemical data such as DPV, CA, and CV allows one to decrease the LoD to 0.022 µM in undiluted human serum. If this technology can be manufactured in POC device format, then such accuracy and precision would allow HU measurements and optimal dosing for patients living in low-resource settings. This work represents an example of multinational and multidisciplinary collaboration, where complex mathematical tools are employed for characterizing and utilizing an electrochemical sensor in a biomedical precision dosing application. Simulated biofluids are more accurate than the usual buffers to address the complex synergy of multicomponent fluids in the electrochemical determination of relevant molecules.

## Figures and Tables

**Figure 1 biomedicines-09-00006-f001:**
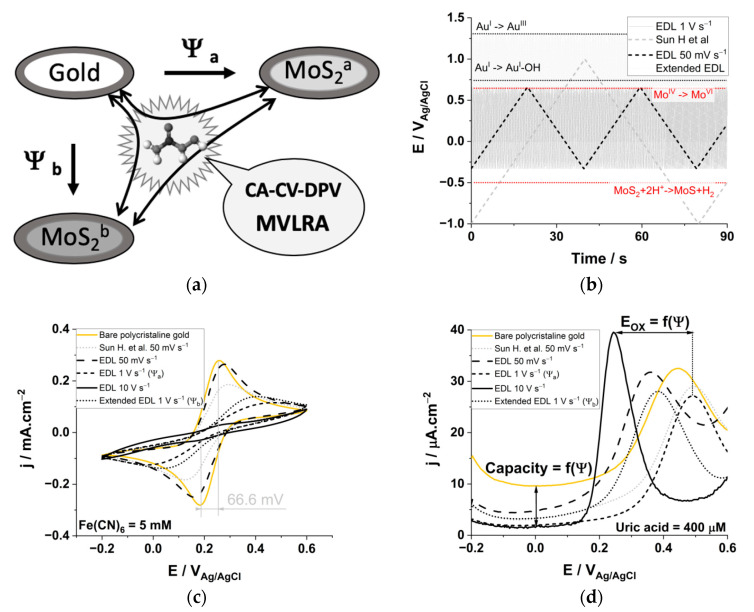
(**a**) Commercial gold electrodes are modified with MoS_2_ using different waveform polarization (Y_n = a,b_) and the concentration of hydroxyurea (HU) is estimated from multiple electro-analytical techniques (ca., chronoamperometry {CA}, cyclic voltammetry {CV} and differential pulse voltammetry {DPV}) using multivariate analytical tools. (**b**) The triangular waveforms used for the electro-deposition of MoS_2_ on the gold electrode, MoS_2_ redox reaction occurs simultaneously for polarization potential below −0.5 V_Ag/AgCl_ and above 0.7 V_Ag/AgCl_ (red dotted lines). For polarization potential > 0.8 V_Ag/AgCl_, gold hydroxylation and oxidation occur (black dotted lines). (**c**) Cyclic voltammograms of the electrodes in the presence of 5 mM ferrocyanide/ferricyanide at 0.1 V s^−1^ in PBS pH 7.4 at a bare electrode (yellow), a MoS_2_ modified electrodes deposited by CV between (Grey dotted, Sun. H et al.) −1/+1 V_Ag/AgCl_, (“EDL”) −0.3/+0.7 V_Ag/AgCl_, (“Extended EDL”) 0/+1.4 V_Ag/AgCl_. (**d**) Differential pulse voltammograms corresponding to the electrochemical signature of 400 mM uric acid in PBS pH 7.4.

**Figure 2 biomedicines-09-00006-f002:**
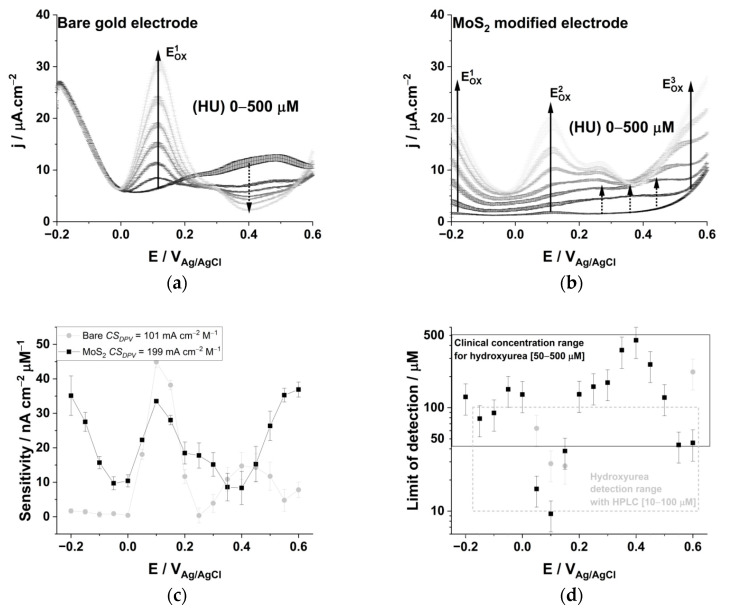
Differential pulse voltammograms at increasing hydroxyurea concentration (HU: 0, 50, 100, 150, 250, 350, and 500 µM) in a phosphate buffer saline solution pH 7.4 recorded on (**a**) a bare polycrystalline gold electrode and (**b**) the MoS_2_ modified gold electrodes. Faradaic (plain black arrow) and capacitive (dotted black arrow) currents synergistically contribute to the dose-response characteristics of the calibration curves. The electrochemical sensor sensitivities at each potential (**c**) is calculated from regression analysis in the concentration range 0–500 µM. The limit of detection for HU (**d**) at the bare (gray) and the MoS_2_ modified electrode (black) are comprised within the biological concentration range.

**Figure 3 biomedicines-09-00006-f003:**
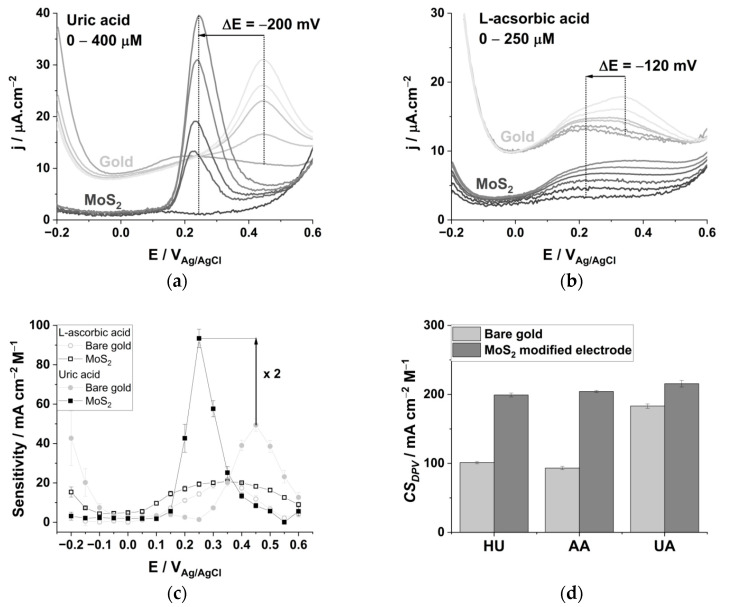
Differential pulse voltammograms (DPV) recorded for (**a**) uric acid and (**b**) L-ascorbic acid at the bare (grey) and the MoS_2_ (black) modified electrodes. (**c**) The sensitivities over the DPV analysis potential range for uric acid (squares) and L-ascorbic acid (circles) at the Bare (grey) and the MoS_2_ (black) modified electrode. (**d**) Combined selectivity of the DPV analysis (*CS_DPV_*) over the full potential range for hydroxyurea (HU), L-ascorbic acid (AA) and uric acid (UA).

**Figure 4 biomedicines-09-00006-f004:**
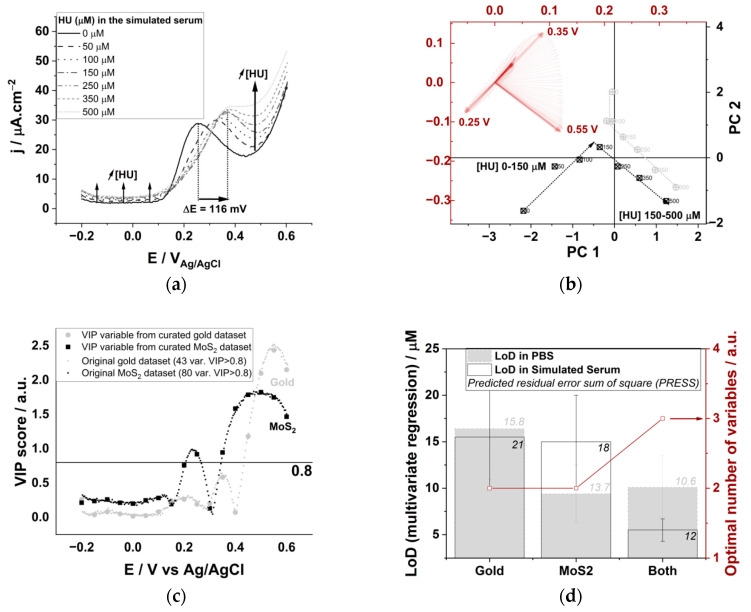
(**a**) Differential pulse voltammograms corresponding to the gradual increase of hydroxyurea (HU: 0, 50, 100, 150, 250, 350, and 500 µM) in a buffered simulated serum composed of 28 redox active molecules in their physiological concentration range acquired with the MoS_2_ modified electrode. (**b**) Principal component analysis of the DPV datasets for HU calibration within its clinical concentration range (0–500 µM) using both the bare (grey) and the MoS_2_ modified (black) electrodes dataset. The loading plot (red) displays the covariance of variables with HU concentration variation. (**c**) Very important variable (VIP) score plot resulting from the partial least square analysis of each DPV dataset for increasing HU concentrations. (**d**) The limit of detection calculated with the multivariate regression model for the isolated and combined electrodes datasets. To increase the optimal number of variables (red) allows for the decrease in the predicted residual sum of square (PRESS, associated to the column plots).

**Figure 5 biomedicines-09-00006-f005:**
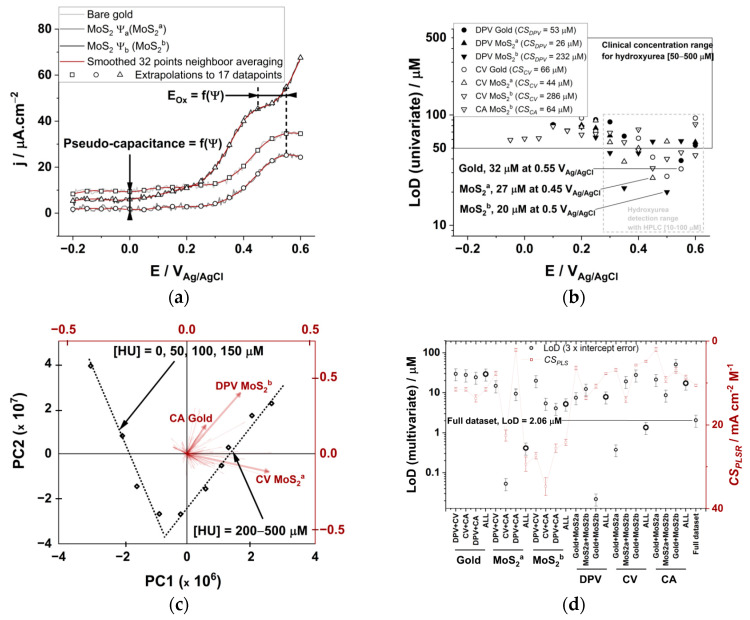
(**a**) DPV analysis of undiluted human serum for the three different electrodes emphasizing the high electrochemical noise recorded upon simultaneous measurements. The variation of the Pseudo-capacitive current and oxidation peak potential is dependent on the electrical waveform used for the electrodeposition of MoS_2_. (**b**) Limit of detection calculated form linear regression analysis of current vs. concentration plots for differential pulse voltammetry (DPV), cyclic voltammetry (CV) and chronoamperometry (CA) for each of the three different electrodes. (**c**) PCA analysis results of the datasets acquired for 10 hydroxyurea concentrations using three electrodes and three electrochemical technics. The loading plot (red) emphasize that each of the technics (DPV, CV, CA) and electrodes constructs (Gold, MoS_2_^a^, MoS_2_^b^) are used to build the multivariate regression model. (**d**) Limit of detection (black) and combined selectivity of the PLSR analysis (CS_PLSR_, red) calculated for different array configurations.

**Figure 6 biomedicines-09-00006-f006:**
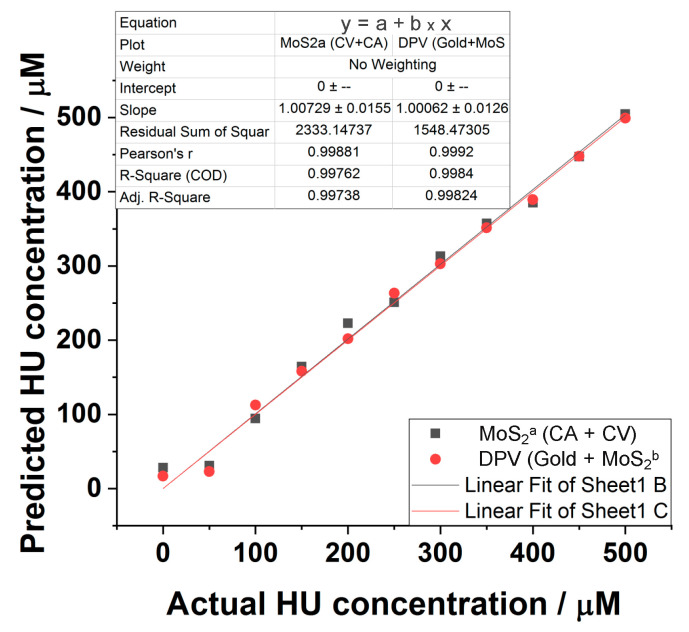
Predicted vs. actual HU concentration plot derived from multivariable linear regression models was calculated from the datasets CV + CA analysis with MoS_2_^a^ (black squares) and DPV analysis of gold + MoS_2_^b^ (red circles).

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
