# Peer review of "Electrochemical Determination of Hydroxyurea in a Complex Biological Matrix Using MoS2-Modified Electrodes and Chemometrics"

_biomedicines, 2020, doi:10.3390/biomedicines9010006_

Round 1

Reviewer 1 Report

In the present article the authors presented the electrochemical determination of hydroxyurea using MoS2 modified electrodes. This work is an example of multidisciplinary collaboration, where complex mathematical tools are employed for characterizing and utilizing of electrochemical sensors in biomedical applications. Generally, the article is well written and organized, the results are presented in a clear manner. However, before acceptance some corrections and improvements are suggested.

The fact is that hydroxyurea detection was also approached before from many years. Various other analytical platforms and methods including colorimetric techniques, high performance liquid chromatography, nuclear magnetic resonance, gas chromatography mass spectrometry, and tandem liquid chromatography mass spectrometry, have been developed to detect and quantify hydroxyurea in biological fluids. The authors are kindly asked to make a discussion (1 -2 paragraphs) related to the use of other analytical platforms in detection of hydroxyurea. This part should be introduced in the Introduction part.

It would have been useful and relevant to have used a confirmation method (like HPLC, for example) to support the obtained results. I will not ask for this like for something mandatory to be realized, but if the authors could do this, they would improve the value of the article.

The novelty/ originality of the study should be clear pointed at the end o of the introduction.

The authors are kindly asked to arrange a bit the Figure 1 (part A) – the elipsoidal of MoS2a is cut, the letter V in the oval below comes out of the circle and the central formula projected on a white background covers the arrows and presents a contrast to the gray background on which it was placed.

I would have liked to see the purity of the substances mentioned in parentheses also. By making a search it was neither easy nor fast to find the mentioned codes where finally I could see the purity.

I consider that to many valuable information was placed in supplementary materials, considering that MDPI journals require publishing in as many details as possible and does not impose any limitation of the article’s length. Please consider moving part of the information in the main text. The figures presented in supplementary material and generally small, so they can be easy combined if the authors don’t want to have large number of figures.

Reviewer 2 Report

The authors reported a study on electrochemical detection of hydroxyurea in a
Complex Biological Matrix Using MoS2-Modified Electrodes. Overall, the ms is well written, with very good quality figures and discussion supported by experimental data. General questions:

1) change Faradic to Faradaic across the ms;

2) how the electrode responds with the variation of pH?

3) It would be good to provide some data about recovery;

4)  how this system compared with others published in literature? it would be good to add a table summarizing the performances of this electrode compared to literature;

Overall, it is a nice contribution.

Round 2

Reviewer 1 Report

The authors answered to all reviewer requests, consequently, I suggest the acceptance of manuscript in the present form.

Kind regards.